# A Prospective Comparison of Bipolar I and II Subjects with and without Comorbid Cannabis Use Disorders from the COGA Dataset

**DOI:** 10.3390/brainsci13081130

**Published:** 2023-07-27

**Authors:** Ulrich W. Preuss, Michie N. Hesselbrock, Victor M. Hesselbrock

**Affiliations:** 1Department of Psychiatry, Psychotherapy and Psychosomatic Medicine, RKH Hospital Ludwigsburg, 71640 Ludwigsburg, Germany; 2Department of Psychiatry, Psychotherapy and Psychosomatics, Martin Luther-University Halle-Wittenberg, 06112 Halle, Germany; 3Department of Psychiatry, University of Connecticut School of Medicine, Farmington, CT 06032, USA; michie.hesselbrock@uconn.edu (M.N.H.); hesselbrock@uchc.edu (V.M.H.)

**Keywords:** cannabis use disorders, bipolar I and II disorders, comorbidity, baseline- and follow-up analysis, alcohol and substance use disorders, PTSD, anxiety disorders

## Abstract

Objective: The comorbidity of alcohol and substance use disorders among persons with bipolar disorder is elevated, as indicated by epidemiological and clinical studies. Following alcohol use, cannabis is the most frequently used and abused illicit substance among bipolar individuals, and such use may lead to comorbid cannabis use disorders (CUD). Previous research indicated that CUDs were related to a more severe course of bipolar disorder and higher rates of other comorbid alcohol and substance use disorders. Few studies, however, have conducted longitudinal research on this comorbidity. The aim of this study is to investigate the influence of CUD on the course of bipolar I and II individuals during a 5-year follow-up. Methods: The characteristics of bipolar disorder, cannabis use disorders, and other alcohol and substance use disorders, as well as comorbid mental disorders, were assessed using a standardized semi-structured interview (SSAGA) at both baseline and the 5-year follow-up. N = 180 bipolar I and II patients were subdivided into groups of with and without comorbid cannabis use disorders (CUD). Results: Of the 77 bipolar I and 103 bipolar II patients, n = 65 (36.1%) had a comorbid diagnosis of any CUD (DSM-IV cannabis abuse or dependence). Comorbid bipolar patients with CUD had higher rates of other substance use disorders and posttraumatic stress disorders, more affective symptoms, and less psychosocial functioning at baseline and at 5-year follow-up. In contrast to previously reported findings, higher rates of anxiety disorders and bipolar disorder complications (e.g., mixed episodes, rapid cycling, and manic or hypomanic episodes) were not found. The effect of CUD on other substance use disorders was confirmed using moderation analyses. Conclusions: A 5-year prospective evaluation of bipolar patients with and without CUD confirmed previous investigations, suggesting that the risk of other substance use disorders is significantly increased in comorbid individuals. CUD has a moderation effect, while no effect was found for other mental disorders. Findings from this study and previous research may be due to the examination of different phenotypes (Cannabis use vs. CUD) and sample variation (family study vs. clinical and epidemiological populations).

## 1. Introduction

Bipolar disorders (BDs) are reported to have a lifetime prevalence of 0.8% for BD-I and 1.1% for BD-II [1]. BDs is a risk factor for both behavioral, (e.g., gambling) [2] and substance use disorders (SUDs) [3,4].

Among these substances, worldwide cannabis is the most widely used illicit drug [5]. Cannabis use (CU) in the United States is reported to be 11.8% in individuals older than 12 years and 10.8% in those individuals aged 26 years and older [5,6]. The CU rate in adults aged 26 and older increased from 7.9% to 10.8% from 2017 to 2020. Cannabis use disorders (CUDs) were diagnosed in 13.5% in 18–25-year-olds and 4.0% in those individuals aged 26+ years. By comparison, among European adults, past-month prevalence of CU increased by 27% from 2010–2019 (from 3.1 to 3.9%), with the most pronounced relative increases observed among 35–64-year-olds [7]. 

A recent review indicates that lifetime cannabis use (CU) was very frequent among persons with BD, ranging from 50–66% using cannabis over their lifetime [8]. This reported rate of lifetime cannabis use (LT-CU) is as much as sevenfold higher in individuals with BD compared to people without BD (71.3%, OR 6.8, CI, 5.41 to 8.52) [9], while cross-sectional prevalence rates of CU varied from 3.3% to approximately 18% [8]. 

Not surprisingly, CUD was also increased among individuals with BD when compared to the general population (7.2% vs. 1.2%, respectively), ranging from 7.2–30% across several studies [9,10,11]. The prevalence rates of CU, cannabis abuse (CA) and cannabis dependence (CD) were found to be higher in BD-I (CD: 23.6%, CA: 9.7%, CUD: 11.8%) versus BD-II (CD: 10.2%, CD: 4.9%, CUD: 5.7%) [12]. 

Interestingly, among individuals diagnosed with CUD, BD comorbidity is lower but still substantial. A meta-analysis found an approximately 10% prevalence rate of BD in individuals with CUD in both community and clinical samples [5,6]. BD-I (8.8–9.0%) is more prevalent than BD-II (0.8–1.5%) in individuals with CUD [13]. Although less common than in CUD, BD is also more common in cannabis users without CUD compared to the general population (2.5%) [14]. In a more recent review and meta-analysis in subjects with cannabis use, the bipolar disorder rate was 9.6 % as indicated by Wave 2 results of the National Epidemiologic Survey on Alcohol and Related Conditions [15]. 

There are certainly significant and complex Interactions between CU, use disorders (CUD) and mood disorders: CU may contribute to the risk for developing psychopathology, which may in turn lead to CU and eventually to CUD. In addition, underlying factors may contribute to both mood disorder psychopathology and CU and CUD [16,17]. 

Previous epidemiological studies [8] indicated that CU is also associated with the worsening of mood disorder symptoms in a dose-dependent manner [10], suggesting that CU and use disorders, like alcohol use disorders [16,17], have a detrimental influence on the development and course of bipolar disorders. 

While there is a difference between CU and use disorders, CU in BD was associated with hypomanic and manic episodes, but not depression [8,18,19]. Individuals with BD and CU displayed increased severities of mania [20], hallucinations, delusions and overall illness [21]. Several studies reported higher rates of complications and more severe bipolar symptoms, including more time in manic and mixed episodes [9], more frequent affective episodes [10], and mixed states [22]. Psychotic symptoms [23] were also more prevalent in BD with CUD.

Further, other SUD also occur more often in individuals with either CUD and BD. Approximately two-thirds of individuals with bipolar disorder and CUD report nicotine dependence and alcohol and drug use disorders. A meta-analysis reported that among SUDs seen in patients with BD, alcohol (42%) and cannabis (20%) were most prevalent, followed by other illicit substances (17%) [5,6]. Other studies confirm that BD with comorbid CU was associated with an increased frequency of SUDs [21]. This is a particularly relevant finding, since co-occurring alcohol and SUDs among individuals with bipolar disorder are negatively associated with on the course of illness (even after adjusting for non-compliance) including a delayed onset of symptomatic recovery when treated [24,25]. The significance of these findings is underscored when considering the heavy burden of disease already associated with bipolar disorder [26,27].

BD patients have a higher prevalence of psychiatric comorbidities with anxiety disorder (24.1%) being most prevalent followed by personality disorders (17.5%) and PTSD (9.7%) [19]. Anxiety disorders, including panic disorders, are reported to occur in BD over the course of disease in high rates ranging from 39% to 55% [28] and PTSD [29]. However, no study has investigated this comorbidity in bipolar subjects with CUD. Comorbid anxiety disorders increase the risk for hospitalization for BD depression but not for BD mania [20]. A higher proportion of BD patients with borderline personality disorders are female and are at higher risk for suicidal behaviors, emphasizing the need for acute inpatient care [19]. Comorbid anxiety disorders increase the odds for hospitalization for BD depression and not for BD mania [28]. 

Few studies, however, used a prospective design (follow-up between 1 to 3 years) to investigate the course of both comorbid disorders. Results from previous research from retrospective and cross-sectional designs indicate a more severe course of comorbid bipolar disorder and CUD [30], more comorbidity [21], more psychopathological symptoms and poorer functional outcome [31]. Importantly, CU was associated with a higher rate of cannabis and other SUDs in an epidemiological prospective study in non-comorbid individuals [32]. 

The aim of the present study was to analyze the clinical course and prognosis of bipolar I and II patients with and without CUDs (DSM-IV Cannabis dependence and Cannabis abuse) using the Collaborative Study on the Genetics of Alcoholism (COGA) sample. Subgroups of bipolar individuals with and without comorbid CUDs were compared over a five-year follow-up time period. Subjects were assessed at baseline regarding their lifetime history of SUD, bipolar disorders and then re-interviewed prospectively after 5 years. In these analyses, both bipolar I and II subjects are included to investigate the lifetime characteristics of CUDs, other alcohol and substance dependence and bipolar disorders retrospectively. Secondly, we examined the course of CUDs, other alcohol and substance dependence, BD and comorbidity with mental and other SUD during the 5-year follow-up period. Thirdly, we analysed the moderating effect of CUD on the risk for developing other alcohol and SUDs or the number of comorbid psychiatric illnesses. 

## 2. Methods

### 2.1. Sample 

The Collaborative Study on the Genetics of Alcoholism (COGA) is a family pedigree investigation which enrolled treatment-seeking alcohol-dependent probands who initially met the DSM-IV for alcohol dependence [33]. Six medical centers in the USA recruited the initial probands plus first-degree family members. The only exclusion criteria include life-threatening medical disorders, repeated intravenous drug use, and an inability to speak English. Written informed consent to participate in the study was obtained from all subjects. Participants and their relatives were interviewed at baseline using the Semi-Structured Assessment for the Genetics of Alcoholism (SSAGA), which focuses on demography, substance use patterns, and the assessment of 17 axis I DSM-IV diagnoses, as well as characteristics of bipolar disorder [16,34]. 

While the SSAGA was developed prior to the publication of the DSM-IV criteria, all criteria symptoms for the DSM-IV diagnosis were assessed ages of onset and remission of symptoms [35]. Only the original probands or comparison subjects, their first-degree relatives, and offspring aged ≤20 years in the participating families were eligible for follow-up. Of all eligible subjects, the follow-up rate was 60% in probands, 65% in family members, and 78% in controls [35].

The interview also assessed past episodes of affective disorders, including depressive and manic episodes and the characteristics of the most severe episode. To receive a DSM-IV bipolar I disorder diagnosis, subjects had to report a lifetime diagnosis of both major depression and mania or any lifetime diagnosis of a manic episode. Individuals who had at least one major depression and hypomanic episode were considered to have bipolar II disorder. 

N = 180 subjects with bipolar I or II disorder were identified. Of these n = 65 (36.1%) had an additional diagnosis of DSM IV CUD (cannabis dependence and cannabis abuse, CUD in 23 of 77 (29.8%) individuals, in bipolar II subjects and 40.8% (42 of 103 individuals in bipolar I subjects). Any CUDs (dependence and abuse) was found in 36.1% of bipolar I and II individuals. 

Subjects with a bipolar II disorder without comorbid CUD but abstinent or with social CU were included into group 1 (n = 54) while group 2 (n = 23) consists of individuals with comorbid bipolar II and CUD diagnoses. Group 3 included subjects with a bipolar I diagnosis without CUD (n = 61) but either abstinence or social CU and group 4 were bipolar I subjects with a comorbid CUD (n = 42). 

The probands and appropriate relatives were re-assessed at a mean of 5.72 years (±1.1 years) after the initial interview. 

### 2.2. Bipolar Patients with and without Any Cannabis Use Disorders

Of the 180 bipolar subjects interviewed at baseline, n = 117 (65.0%) subjects were successfully re-evaluated at follow-up using the SSAGA (Group 1: 62.9%, n = 34; Group 2: 91.3%; n = 21; Group 3: 37.7%, n = 23; group 4: 95.1%; n = 39). There were no statistically significant differences across groups regarding the rate of being re-interviewed, nor were there differences across groups regarding age, gender and number of symptoms during the most severe depressive or manic episode when re-evaluated subjects were compared to those who could not be re-assessed at follow-up. N = 63 individuals did not agree to be re-interviewed, deceased (n = 2) or could not be located due to address change. 

All individuals were interviewed in an outpatient setting usually located in a research lab interview room. All the participants were in euthymic state at time of the interview.

Additional sections of the SSAGA interview were used to determine the age at onset of SUDs (i.e., the age by which three or more criteria were met) and for additional psychiatric diagnoses conditions (e.g., DSM IV anxiety disorders, post-traumatic stress disorders, antisocial personality and conduct disorder). 

Psychopathology and behavior during the interview observed by the interviewer included assessments of appearance, orientation, level of consciousness, memory, mood, and formal thought. Global level of functioning was obtained using the Global Assessment of Functioning (GAF) [33] at baseline and at the follow-up interview. 

Differences across groups were evaluated by using Chi-square statistics for categorical data and one-way analysis of variance (ANOVA) for continuous variables. Scheffé post hoc tests were used to determine significant differences in specific group comparisons. To compare characteristics of continuous variables over time, repeated measurement ANOVA (MANOVA) was used. For post-hoc group comparisons, Scheffé post-hoc tests were employed, when applicable. Group 2 and 4 individuals were compared for characteristics of CUDs while Group 1 and 3 members (per definition no CUD) were skipped from these analyses. 

Moderating effects of any CUD in bipolar patients on number of other substance use (SU) and comorbid mental disorders was computed using SPSS module PROCESS (Ver. 4.1 by Andrew Hayes, http://www.processmacro.org/download.html, accessed 26 July 2023).

## 3. Results

### 3.1. Sample Characteristics 

The sociodemographic characteristics of the four groups are presented in Table 1. At the baseline interview, no differences were noted across groups regarding years of education, employment, ethnicity, marital status, and education. A significantly higher proportion of Group 1 to 3 members compared to Group 4 were female. 

### 3.2. Baseline Analyses

#### 3.2.1. Characteristics of Cannabis Use Disorders 

Individuals with bipolar I and II with Cannabis use disorders (Groups 2 and 4) were compared and presented in Table 2. No significant differences were found regarding age at onset of Cannabis use, number of units used per day, number of DSM-IV criteria endorsed, number of withdrawal symptoms, and other characteristics, e.g., previous treatment or Cannabis-induced affective or other symptoms. 

#### 3.2.2. Comorbidity with Other DSM-IV Mental, Alcohol, and Substance Use Disorders

As shown in Table 3, posttraumatic stress disorder (PTSD) was diagnosed most often in Group 4, while antisocial personality disorder (ASPD) had the highest rate in Group 3. At baseline, comorbid bipolar I and II individuals also had a significantly higher rate of alcohol, cocaine, and stimulant dependence compared to non-comorbid Groups 1 to 3. Group 2 (bipolar II + CUD) had a higher rate of sedative and opioid dependence compared to Groups 1, 3, and 4. 

#### 3.2.3. Characteristics of Manic and Depressive Episodes, Mixed Episodes, Rapid Cycling, and Suicidal Ideation and Behaviors

As expected, bipolar I Individuals with depression and manic episodes reported receiving significantly more often professional help independent of a comorbid Cannabis use disorder. Comorbid bipolar I individuals also reported a higher number of affective symptoms. General psychosocial functioning was higher in bipolar II vs. bipolar I patients, independent of comorbid CUD. As also presented in Table 4, no differences were found across groups regarding the number of mixed episodes, rapid cycling, and suicidal ideation and behavior. 

### 3.3. Follow-Up Analyses 

#### 3.3.1. Characteristics of Cannabis Use Disorders at the 5-Year Follow-Up 

As demonstrated in Table 5, several criteria of CUD were more often found in comorbid bipolar I and bipolar II individuals, including tolerance, withdrawal, and a long time of using and giving up activities. Comorbid bipolar I individuals more often reported a “desire to cut down on use, but could not”, compared to Group 1 to 3 members. 

#### 3.3.2. Comorbid Alcohol and Substance Use, Mental Disorders, and Suicidal Behaviors during a Follow-Up

Comorbid alcohol and substance dependence (including cocaine, stimulant, sedative, and opioid dependence) occur more frequently among comorbid bipolar I and II individuals (see Table 6) and resemble diagnostic rates obtained at baseline. Further, members of Groups 1, 3, and 4 developed depressive episodes more frequently, and bipolar I individuals reported more panic attacks. Non-comorbid bipolar I patients (Group 3) reported a higher number of suicide attempts. Rates of other comorbid mental disorders also mainly replicated those from the baseline assessments. Significant differences were found for higher PTSD rates in comorbid bipolar I subjects across groups. Reports of affective symptoms during the follow-up interview were highest in Group 4 (bipolar I and CUD) individuals. Comorbid bipolar I and II individuals had significantly lower GAF scores than Group 1 and 3 individuals. 

#### 3.3.3. Cannabis Use Disorders as Potential Moderators for Comorbid Alcohol and Substance Use and Mental Disorders

The results of the moderation analysis are presented in Figure 1 and Figure 2. 

The first model (Figure 1) revealed a significant direct and indirect effect of bipolar disorder on a number of other comorbid substance use disorders. The number of Cannabis use disorder criteria met had a significant moderating effect (variance explained 3.7%). In comparison, the direct and indirect effects of bipolar illness and the number of Cannabis disorder criteria on the prevalence of other comorbid psychiatric disorders (model 2, Figure 2) is low (variance explained 4.8%). 

## 4. Medication and Functional and Affective Syndrome Changes over Time of Bipolar Groups

While there was no difference across groups in the rate of subjects receiving medication during their most severe affective episode, subjects were prescribed antidepressants (35%), benzodiazepines (16%), neuroleptics (18%), lithium (9%), anticonvulsants (13%), or a combination of these medications (23%). At the follow-up interview, no differences were found across groups in type and rate of medication for the treatment of the most severe affective episode. Patients were prescribed antidepressants (31%), benzodiazepines (8%), neuroleptics (13%), lithium (35%), and anticonvulsants (22%). At follow-up, 35% of the bipolar subjects in all groups took a combination of these compounds. 

Of the affective symptoms rated by trained SSAGA interviewers, no significant differences between bipolar groups were detected over time (MANOVA F-value: 0.76; df 3; *p*: 0.842). GAF (social functioning) scores significantly improved in Groups 1, 3, and 4 over time; while for Group 2 subjects, a significant decrease in their level of social functioning was observed (MANOVA F-value: 5.17; df 3; *p* < 0.02, see Table 6). 

## 5. Discussion

The lifetime characteristics of CUD, other alcohol- and substance dependence and BDs were examined retrospectively while the course of CUD, other alcohol and substance dependence, bipolar disorder and comorbidity with mental and other SUD were examined during the 5-year follow-up period. We also analysed the possible moderating effect of CUD on the risk for other alcohol and SUD or number of comorbid mental illnesses. 

The primary focus of our analysis of the COGA-data set was to investigate the course of bipolar I and II with and without CUDs (cannabis abuse and dependence). According to the National Survey on Drug use and Health (NSDUH) [6], the rate of CUD in the US general population accounts for 14.2 million individuals (a prevalence rate of approximately 6.0%). In the current sample, the rate of CUD was 36.1% for bipolar I and II individuals. This finding indicates that bipolar disorders are a considerable risk factor for comorbid CUD. Since this is a sample from a family study, the rate of CUD is higher compared to epidemiological studies (Peters et al., 2014) but lower than rates from clinical samples (47.9%) [9]. 

With respect to demography, more females not having a CUD were found in bipolar I and II groups. A previous analysis of this dataset in bipolar individuals with and without alcohol dependence had a similar finding [16]. Bipolar individuals without a CUD also had a higher rate of females, except group 4 (CUD and bipolar I disorder). Therefore, bipolar males may be more prone to develop a CUD compared to females. 

In both the baseline and prospective analyses of characteristics of hypomania and mania, no differences between bipolar II and bipolar I subjects with and without CUD were detected. Previous studies indicated that CU in BD was associated with hypomanic and manic episodes, but not depression [8,18,20]. Also, no significant differences regarding depression episode characteristics were detected across groups with and without CUD. The statistically significant results presented in Table 4 for mania and depression, including seeking help more often from a professional, refer to differences between mania (bipolar I) and hypomania (bipolar II) or various degrees of depression in these subtypes of bipolar disorders. Further, no differences were found in several additional characteristics of bipolar I and II disorders, including rates of rapid cycling, mixed episodes and suicidal ideation and behavior. Previous studies reported higher rates of complications and more severe bipolar symptoms including more time in manic and mixed episodes [9], more frequent affective episodes [10], mixed states [22]. The differing results found in the current analysis may be due to different sample and sampling characteristics. The current sample is a high-risk family study rather than a clinical treatment sample and may therefore show different patterns of comorbidity. 

### 5.1. Comorbidity with Other Mental Disorders Retro- and Prospectively

Comorbidity with psychiatric disorders, including anxiety disorders and DSM-IV antisocial personality disorder, except for posttraumatic stress disorders (PTSD), surprisingly, did not differ across groups. PTSD had a significantly higher rate in comorbid vs. non-comorbid individuals at baseline and comorbid CUD and bipolar I subjects at follow-up. Subjects with PTSD may, therefore, more often use and abuse cannabinoids to cope with their symptoms cf., anxiety, sleep disorders and intrusions. A recent small randomized controlled trial (RCT) using medical cannabis (nabilone) to treat PTSD found that nabilone significantly reduced the severity of PTSD-related nightmares [36]. Participants also reported significant improvements on secondary measures of general well-being and mean global improvements.

This finding, however, parallels the results from a previous analysis of this sample where no influence of alcohol dependence was reported on these characteristics of bipolar disorder [16]. Additional findings such as the number of affective symptoms at baseline interview were most severe in Group 4 and measures of social functioning (GAF) which were lowest in both bipolar I and II subjects with CUD, may reflect a more severe impairment and more mood disorder symptoms in comorbid BD and CUD individuals. 

Comorbid mental disorders in the current analysis included mainly anxiety disorders (panic disorder, PTSD, generalised anxiety disorders, Agoraphobia). Previous biological and genetic studies found a rather weak biological relationship between endogenous cannabinoid system (ECS) functioning [37] or genetics of CUD and anxiety disorders [38], even after controlling for several confounding factors [37]. In comparison, a stronger genetic relationship between affective disorders and ECS has been reported [38]. Recent research also demonstrates that the genetic predisposition for CUD polygenic scores for cannabis phenotypes predicted psychotic disorders independently of other psychiatric disorders [39]. Together these results support the hypothesis that there is little biological or genetic relationship between the ECS and anxiety-related disorders. 

### 5.2. Comorbidity of Any Other Substance Use Disorders 

In comparison, rates of other comorbid substance dependence were significantly increased in comorbid vs. non-comorbid groups at both time-points, except for sedative dependence. Further, no significant difference for any SUD was found comparing comorbid bipolar I vs. bipolar II group members. Since cannabinoids also have some sedative and hypnotic characteristics, users of cannabinoids may therefore abuse other sedatives (e.g., benzodiazepines), less often than other substances such as alcohol, stimulants, or cocaine. However, these results do not support a potential protective effect of CU and abuse from sedative dependence. Individuals with multiple SUDs also include CUD along with alcohol, opioids, stimulants, and cocaine use disorders. Further, the results do not support a ‘gateway’ hypothesis of cannabinoids i.e., CU early in life subsequently increases the risk for developing other alcohol and SUDs. To test this “gateway hypothesis”, a different study design (e.g., longitudinal studies on cannabis users during adolescence and early adulthood) is needed. 

Other studies confirm that BD with comorbid CU is related with increased frequency of SUDs [21,32]. This is a particularly relevant finding, as co-occurring alcohol- and SUDs among individuals with bipolar disorder are associated with negative effects on course of illness (even after adjusting for non-compliance) including a delayed onset of symptomatic recovery when treated [24,25]. The significance of these findings is underscored when considering the heavy burden of disease already associated with bipolar disorder [26,27].

Few studies hitherto used a prospective design (follow-up between 1 to 3 years) to investigate the course of both comorbid disorders [21,30,31]. In contrast to these studies, few results in this sample support the hypothesis that comorbid CUD cause more severe course of bipolar disorder [30]. However, the higher rates of psychopathology and social functioning could be replicated [31]. Higher rates of comorbidity were found for comorbid PTSD in CUD—bipolar I subjects but no more other anxiety disorders. As with a prospective study [21], higher rates of alcohol- and other substance dependence were detected in the current study while rates of manic and hypomanic syndromes did not differ across bipolar Groups 1 to 4. 

Across these types of studies, there are different rates of CU in bipolar disorder. While the current study had high rates of CUDs in bipolar I and II patients (any CUDs in 29.9% of bipolar II and 40.8% in bipolar I subjects). The study of Bahorik et al. [31] reported CU in 27%, Blanco et al. [32] 3.4% in non-bipolar individuals and van Rossum et al. [21] in 12.7% of their samples. However, in this study, compared to previous investigations, data from a highly affected and comorbid sample (CUD and bipolar disorders) are analyzed compared to cannabis use (CU and bipolar disorders) in the previous studies [21,30,31,32]. Thus, individuals in the current sample are affected by multiple substance use and mental comorbidities and have a high rate of comorbid CUD. Further, two previous studies [30,32] used epidemiological samples while clinical patients were enrolled in two other prospective investigations [21,31]. Also, assessment methods differed across prospective studies, using structured interviews and the same sample [30,32], general clinical assessments, self-reports and questionnaires [21] or self-reports, clinical interviews and psychopathology questionnaires [31]. The current sample is from a high-risk genetic family pedigree study. Participants were assessed using structured interviews at two time points four to five years apart which has been reported to have high clinical reliability and validity [34] and may partly explain differing results across studies. 

Thus, the comorbid individuals in the current analysis of the COGA sample may be in a different and more severe stage of their comorbid disorder than individuals with CU and bipolar disorders. In this stage, CUD may be more often co-occur with another SUDs and may be less related to comorbid anxiety disorders. The CUD diagnosis indicates that comorbid individuals have not only a frequent CU but also mental and physical consequences due to the use, withdrawal symptoms and problems to stop or control use. These characteristics may be more strongly associated with the risk of other alcohol- and SUDs rather than anxiety symptoms and disorders. 

The interactions between CUDs and related disorders and mood disorders are certainly complex. CU and CUD may contribute to psychopathology, which may in turn lead to CU. In this sample, however, more potential underlying factors were found for comorbid alcohol and substance use rather than other mental disorders. 

As reported by recent genetic research, there may be a common genetic “addiction” factor which influences all alcohol and SUDs, including CUD, independent of other mental disorders [40,41]. Therefore, the diagnosis of a CUD may increase the liability for developing other alcohol- and SUDs also in these comorbid CUD and bipolar individuals and prospectively not influence the risk for developing an anxiety disorder.

This significant effect of CUD on other SUDs is confirmed by the statistical moderation effects. In this model, a significant indirect effect of CUD (0.44) was found on the relationship between bipolar disorders and number of other alcohol and SUDs. No such effect was detected for CUD on the relationship between bipolar disorders and any other comorbid mental disorders. Thus, comorbid CUD influence significantly the liability for comorbid other SUDs. Certainly, beside genetic factors, other biological and psychosocial mechanisms behind this relationship need to be elucidated in subsequent research. 

From the clinical perspective, comorbid CUD and bipolar patients certainly should be assessed regarding additional alcohol- and SUDs. In treatment, the increased risk for other SUDs should be addressed and specific integrated therapy programs in in- and outpatient settings should be provided [42]. 

There are several limitations to this analysis. First, primarily alcohol-dependent subjects in treatment, their relatives and control families were enrolled into this study. This might explain the rather high rate of alcohol and substance dependence diagnosis in this but may also indicate that a severely comorbid sample with bipolar disorders and several comorbidities in these analyses. Second, several previous investigations included first-episode manic patients to overcome potential bias caused by the number of affective episodes and chronic course of bipolar disorders. The COGA sample included inpatient subjects with and control subjects without alcohol dependence. The enrolees were not first-episode bipolar subjects, compared to other samples [43]. Thus, individuals in a different stage of their disease and more chronic bipolar disorder individuals were recruited. Chronic patients are reported to have a higher rate of previous affective episodes and other comorbidities which in turn increase the likelihood of future affective episodes. When these individuals at different stages of their disease are investigated, it may be more difficult to identify other course predictors and to evaluate the influence of a comorbid disorder on prognosis of bipolar disorder. 

As mentioned above, the sample is neither from a clinical or epidemiological population but rather a family study. Comparison of the study’s results to inpatient samples is therefore limited. Further, many but not all potential comorbidities (e.g., full range of DSM-IV personality disorders) were assessed in this family study. However, the structured clinical interview employed (SSAGA) covered many relevant characteristics of comorbid mental and SUDs in bipolar I and II individuals. In previous studies, different assessment methods were employed, including self-reports, global measures of improvement and clinical questionnaires in samples with bipolar disorders and cannabis use. In this sample, comorbid CUD and bipolar disorder subjects were assessed which may in part explain different results across samples. However, the results from the COGA sample may be representative for a non-clinical but severely affected comorbid CUD and bipolar population where a thorough clinical assessment for additional alcohol- and SUDs is indicated. 

Further, prospective course of comorbid CUD and bipolar disorders should be investigated in available epidemiological or in clinical samples to confirm results from the current analysis. Treatment studies using integrative therapy approaches covering both comorbid disorders should be initiated to meet the needs of these severely affected individuals. 

In the prospective analysis, several additional characteristics of bipolar disorder, including rapid cycling and mixed episode were not traced during the follow-up period. Thus, a higher rate of these potential complications of bipolar disorder in comorbid subjects could not be investigated prospectively. 

A significant number of subjects could not be re-interviewed after 5 years, primarily due to changes in addresses and they could not be contacted (approximately 65%). The characteristics of re-interviewed and not-re-interviewed individuals did not differ significantly across groups. 

Finally, contrasting differences of baseline and follow-up analyses are often observed and may be due to a patient’s ability to recall recent versus past events. In the COGA study, subjects were assessed at baseline regarding their current and lifetime characteristics. At the follow-up interview, they were again assessed for both their current (including most recent events during the follow-up period) period as well as lifetime characteristics. Thus, individuals might recall the more recent events more completely and accurately than more remote events. This potential recall difference underscores the necessity for prospective studies which might provide more accurate data than cross-sectional data collection designs to reduce individual recall bias. 

## Figures and Tables

**Figure 1 brainsci-13-01130-f001:**
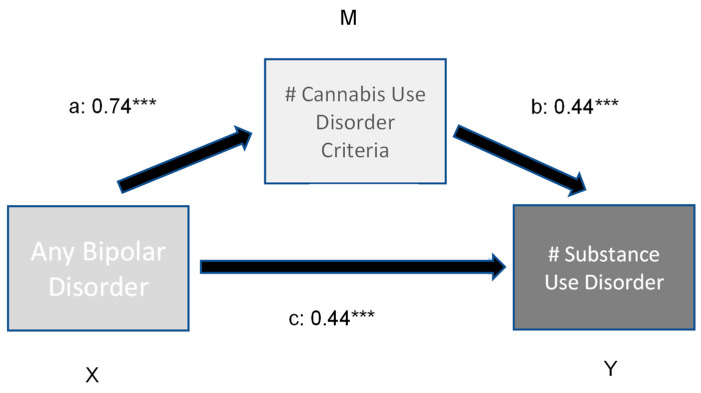
Moderation effect of number of CUD criteria on number of (#) comorbid substance use disorders in bipolar subjects. Indirect effect (X -> Y) 0.24 ***; ***: *p* < 0.001.

**Figure 2 brainsci-13-01130-f002:**
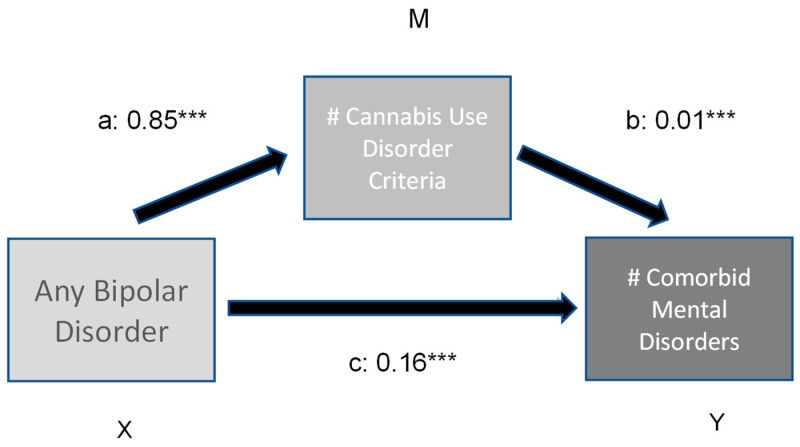
Moderation effect of number of CUD criteria on number of (#) comorbid mental disorders in bipolar subjects. Indirect effect (X -> Y) 0.01 ***; ***: *p* < 0.001.

**Table 1 brainsci-13-01130-t001:** Baseline Demographic Characteristics of Bipolar I and II Subjects Divided by Cannabis Use Disorder Diagnosis.

Variables	GROUP 1 (n = 54) BP2 − CUD	GROUP 2 (n = 23) BP2 + CUD	GROUP 3 (n = 61) BP1 − CUD	GROUP 4 (n = 42) BP1 + CUD	F or χ2 Value	Group Comparisons
Mean (± SD)						
Age (years) Baseline	42.6 ± 12.3	35.6 ± 8.7	43.6 ± 11.9	41.0 ± 7.7	3.10 *	3 vs. 2
Current age (years)	42.4 ± 11.5	39.5 ± 7.5	41.0 ± 7.7	42.2 ± 10.5	0.96	n.s.
Years of education	12.8 ± 2.0	12.7 ± 2.1	12.5 ± 2.4	13.2 ± 2.2	0.66	n.s.
Categorical Variables (%)						
Female Gender	75.9%	65.2%	69.8%	38.2%	14.03 **	1, 2, 3 vs. 4 *
Ethnicity:						
Caucasian	87.0%	91.3%	86.8%	70.6%		
African American	7.4%	8.7%	7.5%	11.8%		
Hispanic	3.7%	0.0%	0.0%	8.8%		
Other	1.9%	0.0%	5.7%	4.3%	10.788	n.s.
Marital Status:						
Married	51.9%	39.1%	35.8%	35.3%		
Widowed	1.9%	0.0%	5.7%	2.9%		
Separated/Divorced	24.1%	26.1%	28.3%	26.5%		
Never Married	27.8%	43.5%	24.5%	29.5%	4.276	n.s.
Unemployed	27.8%	30.4%	49.1%	40.0%	5.980	n.s.
College degree	13.0%	13.0%	13.2%	20.6%	1.218	n.s.

*: *p* < 0.05; **: *p* < 0.01

**Table 2 brainsci-13-01130-t002:** Baseline Alcohol Dependence-related Characteristics and Treatment Histories of Bipolar I and II Subjects Divided by DSM-IV Cannabis Use Disorder Diagnosis.

Cannabis Use Disorder Characteristics (DSM IV) Variables	GROUP 1 (n = 54) BP2 − CUD	GROUP 2 (n = 23) BP2 + CUD	GROUP 3 (n = 61) BP1 − CUD	GROUP 4 (n = 42) BP1 + CUD	F, T or χ2 Value	Group Comparisons 2 vs. 4
Group 2 vs. 4, Mean (±SD)						
Age of onset of Cannabis use	18.70 ± 6.73	15.26 ± 2.5	20.42 ± 9.40	15.94 ± 4.5	0.44	n.s.
Number of units consumed in last 12 m	23.83 ± 52.2	65.86 ± 115.7	105.10 ± 54.2	163.73 ± 51.2	1.77	n.s.
Number of units per day	1.09 ± 0.2	3.52 ± 0.7	1.64 ± 0.49	3.47 ± 1.0	0.04	n.s.
Number of DSM-IV criteria endorsed	0.18 ± 0.61	2.83 ± 1.7	0.11 ± 0.36	3.32 ± 2.2	0.86	
Number of withdrawal symptoms	0	1.65 ± 1.5	0	1.63 ± 1.6	0.006	n.s.
Number of Cannabis side effects	0.45 ± 0.82	0.95 ± 1.4	0.64 ± 1.03	1.03 ± 1.5	0.03	n.s.
Categorical variables (%)						
Ever treatment	1.3%	8.7%	5.3%	11.8%	1.23	n.s.
Cannabis-related characteristics						
Use more than 21x per year	27.5%	100.0%	24.4%	100.0	-	n.s.
Use before age 15	16.2%	100.0%	13.5%	100.0	-	n.s.
Cannabis use consequences						
Depressed	2.6%	34.8%	1.3%	29.4%	0.53	n.s.
Problems with concentration	1.3%	30.4%	2.6%	35.3%	0.06	n.s.
Paranoid or suspicious	1.3%	26.1%	0	29.4%	0.17	n.s.
Social withdrawal	1.3%	60.9%	3.9%	52.9%	0.43	n.s.
Hallucinations	0	13.0%	1.3%	14.9%	0.02	n.s.
Ever hurt under Cannabis	5.3%	95.7%	3.6%	82.4%	3.31	n.s.
Problems with friends or family	2.6%	65.2%	1.3%	47.1%	2.38	n.s.
Use of Cannabis combined with alcohol or other drugs	2.6%	91.3%	6.6%	100.0%	3.92	n.s.
Cannabis withdrawal symptoms						
Nervous, tense, irritable	1.3%	34.8%	1.3%	38.2%	0.07	n.s.
Trouble Sleeping	0	21.7%	1.3%	26.5%	0.16	n.s.
Tremble or Twitch	0	4.3%	0	8.8%	0.56	n.s.
Sweat or Fever	0	8.7%	0	5.9%	0.05	n.s.
Nausea or Vomiting	0	0.0%	0	0.0%	-	n.s.
Stomach aches	0	0.0%	0	2.9%	1.13	n.s.
Changes in appetite	0	26.1%	0	20.6%	0.01	n.s.

m: months.

**Table 3 brainsci-13-01130-t003:** Baseline Cannabis-related Characteristics and Comorbidity with Mental Disorders of Bipolar I and II Subjects Divided by DSM-IV Cannabis Use Disorder Diagnosis.

Cannabis Use Disorder Characteristic (DSM IV) Variables	GROUP 1 (n = 54) BP2 − CUD	GROUP 2 (n = 23) BP2 + CUD	GROUP 3 (n = 61) BP1 − CUD	GROUP 4 (n = 42) BP1 + CUD	F, T or χ2 Value	Group Comparisons
Criteria Cannabis use disorders baseline Group 2 vs. 4						
Great deal of time spent using marijuana		93.8%		81.0%		n.s.
Often wanted to stop or cut down on marijuana		100%		58.4%		n.s.
Tried but was unable to stop or cut down on marijuana		40.0%		50.0%		n.s.
Often used marijuana more frequently or in larger amounts		100%		76.5%		n.s.
Needed larger amounts of marijuana to feel the same effect		88.9%		84.2%		n.s.
Used marijuana to relieve or avoid withdrawal symptoms		75.0%		63.8%		n.s.
Other mental comorbidity:						
Social Phobia	7.1%	20.0%	12.7%	5.9%	3.07	n.s.
Panic disorder	7.1%	5.0%	20.0%	20.5%	6.42	n.s.
Agoraphobia	12.5%	10.0%	12.3%	26.5%	4.51	n.s.
Post-Traumatic Stress Disorder	10.7%	10.0%	12.8%	32.4%	9.07 *	1, 2, 3 vs. 4
Antisocial personality	14.3%	50.0%	18.2%	29.4%	12.13	1, 3, 4 vs. 2
Any other wave I alcohol and substance dependence:						
Alcohol dependence	51.8%	85.0%	59.6%	88.2%	16.74 ***	1, 3 vs. 2, 4
Cocaine Dependence	7.3%	65.0%	21.1%	47.1%	33.39 ***	1, 3 vs. 2, 4
Stimulant Dependence	3.6%	45.0%	10.5%	47.1%	35.93 ***	1, 3, vs. 2, 4
Sedative Dependence	1.9%	13.6%	7.5%	8.8%	15.14 **	1, 2, 4 vs. 3
Opioid Dependence	0.0%	30.0%	12.3%	11.8%	18.04 **	1, 2, 4 vs. 3

*: *p* < 0.05; **: *p* < 0.01; *** *p* < 0.001.

**Table 4 brainsci-13-01130-t004:** Baseline Characteristics of Bipolar Disorders of Bipolar I and II Subjects Divided by Cannabis Use Disorder Diagnosis.

Affective Disorder Characteristics (DSM IV) Variables	GROUP 1 (n = 54) BP2 − CUD	GROUP 2 (n = 23) BP2 + CUD	GROUP 3 (n = 61) BP1 − CUD	GROUP 4 (n = 42) BP1 + CUD	Group Comp.: F or χ2 Value	Group Comparisons
Mania (bipolar I)/Hypomania (bipolar II)						
Mania/Hypomania age of onset	32.08 ± 12.5	25.50 ± 7.9	30.19 ± 12.1	28.64 ± 8.9	1.21	n.s.
Mania/Hypomania number of episodes	12.63 ± 24.2	16.89 ± 24.6	7.64 ± 14.9	15.5 ± 29.8	0.51	n.s.
Mania/Hypomania number of symptoms Mania/Hypomania Treatment for most severe manic/hypomanic episode Seek professional help Medication Hospitalization ECT	4.81 ± 1.8 11.8% 13.3% 0.0% 0.0%	4.93 ± 1.5 11.8% 6.7% 0.0% 0.0%	7.00 ± 2.8 35.3% 40.0% 42.9% 0.0%	7.36 ± 2.4 41.2% 36.8% 57.1% 0.0%	6.03 ** 14.42 1.66 4.47 n.a.	1, 2 vs. 3, 4 * 1, 2 vs. 3, 4 * n.s. n.s.
Depression						
Number of depressive episodes	8.70 ± 19.2	4.39 ± 4.3	10.45 ± 19.9	14.79 ± 25.0	1.46	n.s.
Age of onset depression	21.34 ± 10.6	16.10 ± 7.4	22.78 ± 12.6	20.89 ± 10.3	1.89	n.s.
Any professional treatment most severe episode Seek professional help Medication Hospitalization ECT	9.1% 30.1% 27.6% 20.6% 33.3%	12.5% 11.8% 11.8% 14.7% 0.0%	35.3% 36.6% 36.8% 52.9% 66.7%	42.1% 21.5% 23.7% 11.8% 0.0%	14.20 3.52 1.78 8.28 * 1.81	1, 2 vs. 3, 4 ** n.s. n.s. 1, 2, 4 vs. 3 n.s.
Number of symptoms most severe depressive episode	8.82 ± 1.8	7.86 ± 2.4	8.07 ± 2.2	8.61 ± 1.9	1.13	n.s.
Affective symptoms at baseline interview (Interviewer rating)	1.41 ± 0.4	1.5 ± 0.5	1.38 ± 0.6	2.21 ± 0.9	4.32 **	1, 2, 3 vs. 4
GAF baseline (Interviewer rating)	66.02 ± 18.8	73.78 ± 8.7	58.59 ± 20.5	58.88 ± 12.7	5.61 **	1, 3, 4 vs. 2
Other characteristics						
Any mixed episodes	28.0%	20.0%	28.0%	24.0%	2.40	n.s.
Any rapid cycling	26.5%	20.6%	29.4%	23.5%	0.77	n.s.
Suicide ideation	24.8%	14.3%	35.3%	25.6%	7.67	n.s.
Suicide attempts	26.3%	7.0%	35.1%	31.6%	5.07	n.s.
Number of suicide attempts	4.20 ± 9.5	2.25 ± 1.9	1.85 ± 2.9	3.1 ± 3.4	0.53	n.s.
Age at first suicide attempt	27.07 ± 12.9	20.50 ± 5.2	23.45 ± 9.4	25.17 ± 12.2	0.51	n.s.

*: *p* < 0.05; **: *p* < 0.01; GAF: global assessment of functioning.

**Table 5 brainsci-13-01130-t005:** Characteristics of Cannabis Use during 5-year Follow-Up in Bipolar I and II Subjects Divided by Cannabis Use Disorder Diagnosis.

Variables	GROUP 1 (n = 34) BP2 − CUD	GROUP 2 (n = 21) BP2 + CUD	GROUP 3 (n = 23) BP1 − CUD	GROUP 4 (n = 39) BP1 + CUD	Group Comp.; F, or χ2 Value	Group Comparisons
Comparison of groups						
Cannabis-related characteristics during follow-up period						
Cannabis use	16.4%	16.4%	34.3%	32.8%	5.25	n.s.
Craving	5.9%	17.6%	0.0%	76.5%	6.93	n.s.
Desire to cut down	0.0%	36.4%	0.0%	63.6%	3.83	n.s.
Used more than intended	3.7%	14.8%	3.7%	77.8%	12.58 **	1, 2, 3 vs. 4
Wanted to stop but could not	0.0%	36.4%	0.0%	63.3%	3.83	n.s.
Tolerance	0.0%	29.4%	5.9%	64.7%	10.14 *	1, 3 vs. 2, 4
Withdrawal	3.7%	25.9%	3.7%	66.7%	45.72 ***	1, 3 vs. 2, 4
A long time using	2.1%	37.5%	2.1%	58.3%	22.67 ***	1, 3 vs. 2, 4
Treatment	3.4%	6.9%	37.9%	51.7%	1.49	n.s.
Continued despite knowledge	0.0%	42.4%	3.0%	54.5%	3.10	n.s.
Given up activities	0.0%	34.4%	3.1%	62.5%	11.35 *	1, 3 vs. 2, 4
Treatment	0.0%	22.2%	0.0%	77.8%	2.08	n.s.

*: *p* < 0.05; **: *p* < 0.01; *** *p* < 0.001.

**Table 6 brainsci-13-01130-t006:** Characteristics of Alcohol and Substance Dependence during 5-year Follow-Up in Bipolar I and II Subjects Divided by Cannabis Use Disorder Diagnosis.

	GROUP 1 (n = 34) BP2 − CUD	GROUP 2 (n = 21) BP2 + CUD	GROUP 3 (n = 23) BP1 − CUD	GROUP 4 (n = 39) BP1 + CUD	Group Comp.; F, or χ2 Value	Group Comparisons
Alcohol and Substance dependence at Wave II						
Alcohol dependence Wave II	46.4%	80.0%	45.6%	82.4%	18.84 ***	1, 3 vs. 2, 4
Cocaine dependence Wave II	7.3%	65.0%	21.1%	41.7%	33.39 ***	1, 3 vs. 2, 4
Stimulant dependence Wave II	3.6%	40.0%	10.5%	38.2%	26.12 ***	1, 3 vs. 2, 4
Sedative dependence Wave II	0.0%	30.0%	8.8%	8.8%	17.13 ***	1, 3, 4 vs. 2
Opioids dependence Wave II	0.0%	15.0%	5.3%	23.5%	16.96 ***	1, 3 vs. 2, 4
Comorbidity with mental disorders and suicidal behavior						
During follow-up depression	27.4%	11.6%	36.3%	24.7%	12.90 **	1, 3, 4 vs. 2
During follow-up dysthymia	26.5%	11.8%	29.4%	32.4%	0.94	n.s.
During follow-up manic episode	4.7%	1.2%	55.3%	38.8%	0.71	n.s.
During follow-up hypomanic episode	79.4%	20.6%	-	-	0.26	n.s.
During follow-up panic attacks	25.0%	8.3%	36.1%	30.6%	8.49 *	1, 2, vs. 3, 4 *
During follow-up suicide attempt	16.7%	0.0%	50.0%	33.3%	8.48 *	1, 2, 4 vs. 3 *
During follow-up suicidal ideations	21.3%	9.3%	38.7%	30.7%	6.01	n.s.
ASPD Wave II	14.3%	35.0%	20.0%	26.5%	4.49	n.s.
PTSD Wave II	8.9%	10.0%	10.9%	32.4%	11.01 *	1, 2, 3 vs. 4
Panic disorder Wave II	3.6%	5.0%	5.3%	2.9%	0.38	n.s.
Agoraphobia Wave II	10.7%	10.0%	10.5%	23.5%	4.00	n.s.
Social Phobia Wave II	7.1%	20.0%	12.7%	5.9%	3.07	n.s.
Affective symptoms at follow-up interview (interviewer rating)	1.33 ± 0.5	1.20 ± 0.4	1.00 ± 0.4	2.18 ± 0.9	4.81 *	1, 2, 3 vs. 4 *
GAF follow-up (interviewer rating)	74.82 ± 13.5	68.81 ± 13.5	75.83 ± 9.6	66.67 ± 13.1	3.83 *	1, 3 vs. 2, 4 *

*: *p* < 0.05; **: *p* < 0.01; *** *p* < 0.001.

## Data Availability

COGA Collaboration.

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
