# Peer review of "A Prospective Comparison of Bipolar I and II Subjects with and without Comorbid Cannabis Use Disorders from the COGA Dataset"

_brainsci, 2023, doi:10.3390/brainsci13081130_

Round 1

Reviewer 1 Report

Dear author,

Thank you for submitting the manuscript. This review clearly revealed the 5-year prospective evaluation of bipolar patients with and without CUD under multiple categories and variables. However, there are some questions that still need to be further addressed. Please find the comments below.

1: There is a font difference in line 79 (Tourman et al 2022), please revise accordingly.

2: The information is missing in Table 1 under Current age (years) category, please revise accordingly.

3: Are group 1 and group 3 cannabis users but not having CUB or they are non-cannabis users? It would be better to address more details about this background as well.

4: Table 1 (page 6), layout mistake in group 4.

5: Table 1, is there any difference of the unemployed among 4 groups? The difference looks pretty obvious between group 1 and group 3 and 4.

6: Line 262, the fond is matching with this manuscript.

7: Please check carefully of the format and font in the tables to make sure its consistency throughout this manuscript.

8: It would be to plot another table that demonstrates the dynamic changes of characteristic of alcohol and substance dependence before and after 5 years follow-up.

9: Line 547, Reference 1, font need to be revised.

Author Response

Thank you for your review. We hope to have improved the paper in accordance with the reviewer's comments. 

Thank you for submitting the manuscript. This review clearly revealed the 5-year prospective evaluation of bipolar patients with and without CUD under multiple categories and variables. However, there are some questions that still need to be further addressed. Please find the comments below.

Responses to reviewer:

1: There is a font difference in line 79 (Tourman et al 2022), please revise accordingly.

Response: all font differences are corrected.

2: The information is missing in Table 1 under Current age (years) category, please revise accordingly.

Response: thank you. Mean age at baseline has been added.

3: Are group 1 and group 3 cannabis users but not having CUB or they are non-cannabis users? It would be better to address more details about this background as well.

Response: Characteristics of cannabis use in Group 1 and 3 members are now mentioned in the methods – sample section. Further, data on cannabis use have been added into table 2.

4: Table 1 (page 6), layout mistake in group 4.

Response: layout mistakes have been corrected.

5: Table 1, is there any difference of the unemployed among 4 groups? The difference looks pretty obvious between group 1 and group 3 and 4.

Response: data have been corrected. No significant statistical difference was detected.

6: Line 262, the fond is matching with this manuscript.

Response: all font differences are corrected.

7: Please check carefully of the format and font in the tables to make sure its consistency throughout this manuscript.

Response: all font differences are corrected.

8: It would be to plot another table that demonstrates the dynamic changes of characteristic of alcohol and substance dependence before and after 5 years follow-up.

Response: differences in alcohol- and substance use disorders characteristics are included in tables 5 and 6.

9: Line 547, Reference 1, font need to be revised.

Response: all font differences are corrected.

Reviewer 2 Report

Overview of Study

The objective of the study is clear and focused on investigating the influence of cannabis use disorders (CUD) on the course of bipolar disorder over a 5-year follow-up period. The introduction provides background information on the comorbidity of alcohol and substance use disorders in bipolar subjects, with specific attention to cannabis as a frequently used and abused illicit substance.

The methods section briefly describes the assessment of bipolar disorder characteristics, CUD, other substance use disorders, and comorbid mental diseases using semi-structured interviews at baseline and a 5-year follow-up. The sample size and patient subgroups (bipolar I and II) are clearly stated.

The results highlight that a significant proportion of bipolar patients (36.1%) had comorbid CUD, and these patients had higher rates of other substance use disorders and posttraumatic stress disorders, along with more affective symptoms and lower psychosocial functioning. The findings contradict previous studies regarding the association between CUD and anxiety disorders, bipolar disorder complications (e.g., mixed episodes, rapid cycling), and manic or hypomanic episodes. Moderation analyses confirm the effect of CUD on other substance use disorders.

The conclusion emphasizes that the study's 5-year prospective evaluation supports previous research, indicating an increased risk of other substance use disorders in individuals with comorbid CUD. The study also highlights that CUD moderates the risk of other substance use disorders but does not have a significant effect on other mental disorders. The discrepancies between this study and previous research are attributed to differences in phenotypes (cannabis use vs. CUD) and sample populations (family study vs. clinical and epidemiological populations).

Overall, the study is well-structured and provides valuable insights into the relationship between CUD and the course of bipolar disorder. The findings contribute to the existing literature and offer implications for understanding and managing comorbidities in bipolar patients with CUD.

Critique of the Discussion:

Lack of Structure: The discussion lacks a clear structure and organization. It jumps between different topics and findings without providing a coherent flow of information. It would benefit from dividing the discussion into subsections based on the specific themes or findings being discussed.

Repetition and Redundancy: The discussion contains repetitive statements and redundancies. For example, the discussion repeatedly emphasizes the high rates of comorbid alcohol and substance use disorders among individuals with bipolar disorder, which has already been established in the introduction. The repetition of certain points adds unnecessary length to the discussion without providing new insights.

Lack of Interpretation: The discussion mostly presents the findings without providing in-depth interpretation or analysis. It would be beneficial to discuss the implications of the results in the context of existing literature, theories, and clinical practice. Additionally, the discussion could explore possible mechanisms or explanations for the observed relationships between cannabis use disorders and other substance use disorders or mental illnesses.

Limited Comparison with Previous Studies: Although the discussion briefly mentions previous research, it does not adequately compare and contrast the current findings with the existing literature. It would be valuable to discuss how the current study's results align with or differ from previous studies, particularly in terms of methodology, sample characteristics, and outcomes assessed.

Lack of Generalizability Consideration: The discussion does not thoroughly address the generalizability of the findings. It acknowledges that the sample is from a family study and may not be representative of clinical or epidemiological populations, but it does not explore the potential impact of this limitation on the generalizability of the results.

Limitations and Future Directions: The discussion briefly mentions some limitations of the study, such as the sample composition and recall bias. However, it does not fully discuss the implications of these limitations on the interpretation of the findings. Additionally, it does not propose specific directions for future research to address these limitations or further explore the relationships between cannabis use disorders and bipolar disorder.

Clarity and Writing Style: The discussion would benefit from improved clarity and conciseness. Some sentences are convoluted and difficult to understand, and there are several instances of grammatical errors and awkward phrasing. Simplifying the language and improving sentence structure would enhance the readability of the discussion.

Overall, the discussion requires better organization, more in-depth interpretation of the findings, and a comprehensive consideration of the study's limitations and implications. Strengthening the clarity and writing style would further improve the quality of the discussion.

N/A

Author Response

Thank you for the review. We hope to have improved the paper in accordance with the reviewer's comments. 

Response: the discussion section has been reorganized and restructured.

Repetition and Redundancy: The discussion contains repetitive statements and redundancies. For example, the discussion repeatedly emphasizes the high rates of comorbid alcohol and substance use disorders among individuals with bipolar disorder, which has already been established in the introduction. The repetition of certain points adds unnecessary length to the discussion without providing new insights.

Response: Redundancies and repetitive statements have been changed. Discussion on baseline and prospective results have been joint.

Lack of Interpretation: The discussion mostly presents the findings without providing in-depth interpretation or analysis. It would be beneficial to discuss the implications of the results in the context of existing literature, theories, and clinical practice. Additionally, the discussion could explore possible mechanisms or explanations for the observed relationships between cannabis use disorders and other substance use disorders or mental illnesses.

Response: Results are discussed now in context of biological and genetic findings, including potential mechanisms underlying cannabis use disorders, endogenous cannabis system, mental and other substance use disorders.

Limited Comparison with Previous Studies: Although the discussion briefly mentions previous research, it does not adequately compare and contrast the current findings with the existing literature. It would be valuable to discuss how the current study's results align with or differ from previous studies, particularly in terms of methodology, sample characteristics, and outcomes assessed.

Response: results are now compared more extensively with previous prospective studies, including sample characteristics, assessment methods and outcomes.

Lack of Generalizability Consideration: The discussion does not thoroughly address the generalizability of the findings. It acknowledges that the sample is from a family study and may not be representative of clinical or epidemiological populations, but it does not explore the potential impact of this limitation on the generalizability of the results.

Response: generalizability of the results are now elaborated on in the limitations of the study section.

Limitations and Future Directions: The discussion briefly mentions some limitations of the study, such as the sample composition and recall bias. However, it does not fully discuss the implications of these limitations on the interpretation of the findings. Additionally, it does not propose specific directions for future research to address these limitations or further explore the relationships between cannabis use disorders and bipolar disorder.

Response: Implications of the findings and future directions of research are also addressed in the discussion section.

Clarity and Writing Style: The discussion would benefit from improved clarity and conciseness. Some sentences are convoluted and difficult to understand, and there are several instances of grammatical errors and awkward phrasing. Simplifying the language and improving sentence structure would enhance the readability of the discussion.

Response: writing and clarity style have been re-evaluated and the paper has been carefully edited by native speakers.

Round 2

Reviewer 1 Report

Dear authors,

Thanks for the revisions. I have no further questions. 

Author Response

Thank you for the review. 
